# How do trends in mortality inequalities by deprivation and education in Scotland and England & Wales compare? A repeat cross-sectional study

Gerry McCartney,[1] Frank Popham,[2] Srinivasa Vittal Katikireddi,[2] David Walsh,[3] Lauren Schofield[4]

[1]Public Health Science Directorate, NHS Health Scotland, Glasgow, Scotland
[2]CSO/MRC Social and Public Health Sciences Unit, Institute of Health and Wellbeing, University of Glasgow, Glasgow, Scotland
[3]Glasgow Centre for Population Health, Glasgow, Scotland
[4]Public Health Intelligence, NHS National Services Scotland, Edinburgh, Scotland

**Correspondence to**
Dr Gerry McCartney;
gmccartney@nhs.net

## ABSTRACT

**Objective** To compare the trends in mortality inequalities by educational attainment with trends using area deprivation.

**Setting** Scotland and England & Wales (E&W).

**Participants** All people resident in Scotland and E&W between 1981 and 2011 aged 35–79 years.

**Primary outcome measures** Absolute inequalities (measured using the Slope Index of Inequality (SII)) and relative inequalities (measured using the Relative Index of Inequality (RII)) in all-cause mortality.

**Results** Relative inequalities in mortality by area deprivation have consistently increased for men and women in Scotland and E&W between 1981–1983 and 2010–2012. Absolute inequalities increased for men and women in Scotland, and for women in E&W, between 1981–1983 and 2000–2002 before subsequently falling. For men in E&W, absolute inequalities were more stable until 2000–2002 before a subsequent decline. Both absolute and relative inequalities were consistently higher in men and in Scotland. These trends contrast markedly with the reported declines in mortality inequalities by educational attainment and apparent improvement of Scotland's inequalities with those in E&W.

**Conclusions** Trends in health inequalities differ when assessed using different measures of socioeconomic status, reflecting either genuinely variable trends in relation to different aspects of social stratification or varying error or bias. There are particular issues with the educational attainment data in Great Britain prior to 2001 that make these education-based estimates less certain.

## BACKGROUND

Health inequalities are systematic, avoidable and unjust differences in health outcomes[1] representing substantial unnecessary mortality and morbidity across countries.[2][3] There has been a renewed policy focus on reducing health inequalities in recent years which has in turn led to debate about what should constitute success.[4]

Measures of population health tend to be seen as a relative phenomenon in that comparisons between populations are what

### Strengths and limitations of this study

► This study uses data on the whole population over time stratified into 10 groups to calculate mortality inequalities and therefore avoids sampling bias and aggregation of heterogeneous groups.
► Unlike educational attainment, the use of area deprivation as a measure of socioeconomic position avoids difficulties relating to compositional change in the population over time.
► Our area-based measure of socioeconomic position applies a mean deprivation score to all people living within a postcode sector which will misclassify some individuals.
► The measures used to calculate the Carstairs Deprivation Index are a pragmatic collection of indicators drawn from the census and do not fully capture the lived reality of deprivation and may be insensitive to changes in that experience over time.

help to contextualise whether a particular population health outcome is (relatively) good or bad. Health inequalities measures are similar in this way—it is how a health inequality measure for a population compares to those for other populations, including itself over time, that allows progress to be assessed and decisions made on priorities for action.[5] However, the health inequality measured for a single population can also be presented as either an absolute or relative difference. The interpretation of the trend in health inequalities within a country and differences in health inequality between countries can differ by whether assessed as absolute or relative difference.

Mackenbach *et al* have argued from a pragmatic position that reduction in absolute difference should be considered a success because reductions in relative differences are more difficult to achieve when health is improving.[6][7] However, relative measures of inequalities in mortality improved in both

Great Britain and the USA using different measures prior to the changing political context in both countries from the 1980s, despite there being improvements in the mean health of the populations at this time.[8 9] This suggests that it is possible and feasible to expect both absolute and relative measures of health inequalities to improve simultaneously given a conducive policy context.

In addition to relative and absolute measures of health inequality, there are also a variety of means of measuring socioeconomic position (eg, by income, social class, area deprivation, educational attainment) and many different health outcomes which are of interest (eg, all-cause mortality, well-being, cause-specific morbidity, self-rated health). All of these options have merit from a theoretical perspective and are likely to represent the operation of different social processes.[10–12] Monitoring trends in health inequalities using relative and absolute measures, for a number of different social ranking and outcome measures, is therefore likely to be important to provide a comprehensive understanding of health inequality trends.[13] However, it is common for there to be either no available data for many measures or for the data to have substantial limitations, leading to pragmatic decisions being made on the preferred measures for monitoring health inequality trends within and between countries.

Recent observed declines in absolute measures of inequality for mortality rates between populations ranked by educational attainment between 1990 and 2010 in Europe have been interpreted as demonstrating substantial progress in achieving a reduction in health inequality.[7] However, these reductions need to be interpreted alongside other analyses because some of the data are not nationally representative (such as those drawn from cities) and because of problems with the educational attainment data on which some of the analyses are based.[14 15] Cautious interpretation of these data is necessary, as the priority given by policymakers to reducing health inequalities may be dependent on their understanding of these trends. Furthermore, the perceived success or failure of previous policy decisions may also be judged using these data—therefore, guiding future decision-making.

To assess whether the improvements in absolute inequalities reported by Mackenbach *et al* for mortality inequalities by educational attainment[7] are also seen using other measures of social status, we compare the health inequalities trends for the same population strata (men and women aged 35–79 years) resident in Scotland and England & Wales (E&W) using area deprivation. We then discuss the strengths and limitations of the different measures before reflecting on the implications for policy.

## METHODS

We report our methods here in line with the Guidelines for Accurate and Transparent Health Estimates Reporting statement.[16]

## Data

Individual mortality records were obtained from the National Records of Scotland (NRS) and the Office for National Statistics (ONS). Data were for residents of Scotland, and E&W only. Mortality records were linked to census wards (for E&W) and postcode sectors (for Scotland) of usual residence to allow allocation of area deprivation.

We extracted data on all deaths among those aged 35–79 years (to match the age strata used in Mackenbach *et al*) in 5-year age bands, separately for men and women, for 3 years close to each census year (1981–1983,* 1990–1992, 2000–2002, 2010–2012) to provide more stable figures. The only exclusions were for those without a valid postcode (which tended to be for non-residents).

### Socioeconomic exposure variable

The Carstairs Deprivation Score is derived from four census variables calculated for small geographical areas: the proportion of economically active males seeking work; the proportion of people living in private households at a density of more than one person per room; the proportion of economically active males in occupational social class four or five on the Registrar General's categorisation and the proportion of all persons in private households without access to a car or van.[17] Using the mean and SD of this derived deprivation variable, postcode sectors (for Scotland, mean population 5600 in 2011) and census wards (for England, mean population 6540 in 2011) were ranked and divided into 10ths (population weighted such that each decile had an equal population size) and each mortality record allocated a corresponding area deprivation score (1–10). The postcode sectors/wards and their deprivation scores were calculated according to the geographies and census data available at each time point. The deprivation scores for Scotland and E&W were calculated separately (ie, the Scottish population was divided into 10 equally sized units according to the deprivation score, and the E&W population was similarly divided into 10 equally sized units).

### Statistical analysis

We used the revised census population estimates for each of the census years published by NRS and the ONS as our denominator populations, multiplying the denominator by three to match the numerators (which were 3 years of deaths). As population undercounts were a feature of the 1981 and 1991 censuses,[18] census populations for both Scotland and E&W were constrained to the revised midyear population estimates for these census years by age group and sex.

We standardised the mortality rates using the 1976 European standard population for each 10th of the population ranked by Carstairs deprivation, for comparability with Mackenbach *et al*.

---

* 1981–1983 was used instead of 1980–1982 because English deaths data were not available for 1980.

We calculated mortality rates for each population, sex and time period using Poisson regression, in Stata, with population as the offset and used the margins command to standardise rates.[19] The Relative Indices of Inequality (RIIs) were also obtained from Poisson models. For the RIIs, we calculated the midpoint of the cumulative distribution of people in each decile based on the midyear population of people aged 35–79 years for each country, sex and year combination. Following Mackenbach et al, Slope Indices of Inequality (SIIs) were calculated as 2*ASMR*(RII–1)/(RII+1), where ASMR is the age-standardised mortality rate for that country, sex, year combination. The SII CIs were calculated from the CIs for the RII and ASMR.

To compare with published mortality inequalities based on educational attainment, we interpolated the trends between census years, using log-linear interpolation for the RIIs and linear interpolation for the SIIs, to obtain estimates for the same years for which Mackenbach et al provide data (1993, 1993/1994, 2007/2008 and 2008). SII and RII estimates were published by Mackenbach using educational attainment data for Scotland but not E&W (as the latter were based on only two categories of educational attainment). The published time periods for the trends in inequalities (RII and SII for Scotland; rate ratio differences and absolute differences for Scotland and E&W) by educational attainment were 1991–1995 to 2006–2010 for Scotland, and 1991–1996 to 2006–2009 for E&W.[7]

Simple percentage changes for Scotland and E&W for men and women and for changes in RII and SII changes were then calculated using our Carstairs deprivation-based estimates and compared with the published estimates by Mackenbach et al.

## RESULTS

Mortality rates for those aged 35–79 years were consistently higher in the more deprived areas (for men and women, and in Scotland and E&W), with a linear relationship between deprivation and mortality (online supplementary table 1 and supplementary figures 1–4). Relative inequalities increased steadily between 1981–1983 and 2010–2012 in all groups, with RIIs consistently higher in men and in Scotland (online supplementary tables 2 and 3). The trends for absolute inequalities across the population were more mixed: for men and women in Scotland the SIIs increased between 1981–1983 and 2000–2002 before falling; while in E&W the trends were stable for men until 2000–2002 before a subsequent decline and increased for women until 2000–2002 before a subsequent decline. The SII figures were again consistently higher for men and in Scotland (table 1 and figure 1).

These trends in mortality inequalities by Carstairs deprivation are markedly different from those reported by educational attainment. Between the early 1990s and late 2000s, the reported RIIs for mortality inequalities by educational attainment[7] declined in Scotland by 45% and 20% for men and women, respectively. In contrast, the RIIs by Carstairs deprivation increased by 31% and 24% for men and women. The reported declines in educational attainment by SII for Scotland were even larger at 64% and 47%, respectively,[7] but again these were markedly different from the observed stability among men (5% reduction) and small increase of 1% for women using Carstairs deprivation for the same period (figure 2, table 2).

There was no trend in RII or SII for E&W by educational attainment available for direct comparison, but the reported trends in relative differences were almost zero for educational attainment, whereas our analyses showed increases of 38% and 23% for relative changes in Carstairs deprivation using the RII. The trends in absolute differences by educational attainment reported for E&W were –36% and –24% for men and women.[7] The equivalent

**Table 1** RII and SII by Carstairs deprivation decile over time (men and women aged 35–79 years)

| | Males | | Females | |
|---|---|---|---|---|
| Year | RII (95% CI) | SII per 100 000 per year (95% CI) | RII (95% CI) | SII per 100 000 per year (95% CI) |
| England and Wales | | | | |
| 1981–1983 | 1.61 (1.59 to 1.62) | 768 (753 to 784) | 1.43 (1.42 to 1.45) | 325 (315 to 335) |
| 1990–1992 | 1.76 (1.74 to 1.78) | 748 (734 to 762) | 1.58 (1.56 to 1.60) | 356 (347 to 366) |
| 2000–2002 | 2.12 (2.10 to 2.14) | 758 (746 to 770) | 1.86 (1.84 to 1.88) | 395 (387 to 404) |
| 2010–2012 | 2.30 (2.27 to 2.32) | 608 (598 to 617) | 2.10 (2.08 to 2.13) | 362 (355 to 370) |
| Scotland | | | | |
| 1981–1983 | 1.59 (1.55 to 1.64) | 887 (833 to 941) | 1.54 (1.50 to 1.59) | 471 (437 to 506) |
| 1990–1992 | 1.85 (1.79 to 1.90) | 962 (913 to 1012) | 1.69 (1.64 to 1.74) | 497 (464 to 531) |
| 2000–2002 | 2.34 (2.27 to 2.41) | 1062 (1019 to 1105) | 1.98 (1.91 to 2.05) | 530 (500 to 560) |
| 2010–2012 | 2.64 (2.55 to 2.72) | 879 (845 to 912) | 2.24 (2.16 to 2.32) | 500 (474 to 526) |

RII, Relative Index of Inequality; SII, Slope Index of Inequality.

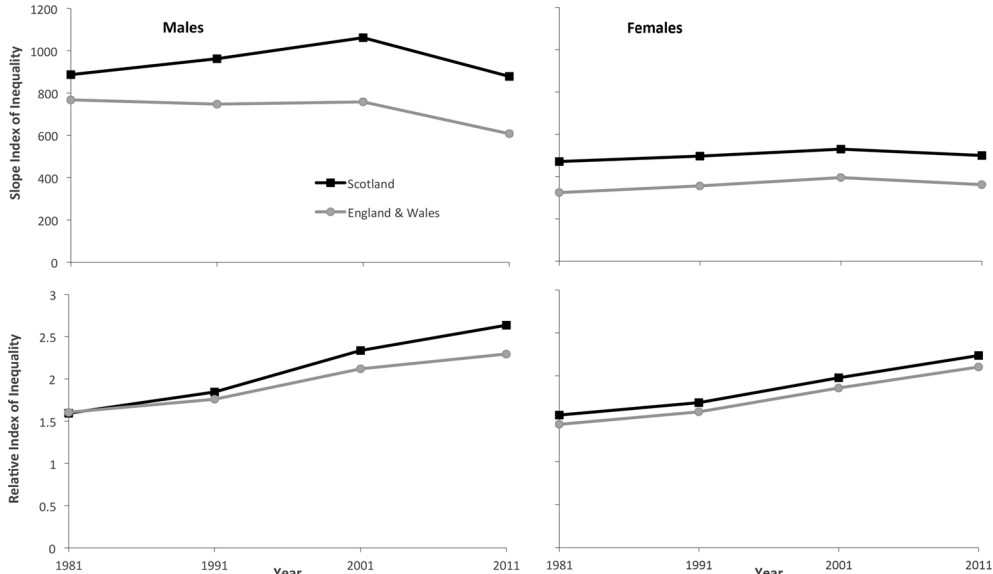

**Figure 1** Trends in SII and RII by Carstairs area deprivation (1982–2011, Scotland and England and Wales, men and women). RII, Relative Index of Inequality; SII, Slope Index of Inequality.

changes in absolute inequalities (using the SII) for E&W using Carstairs deprivation were −12% for men and an increase of 2% for women (supplementary tables 3–5).

## DISCUSSION
### Main results
Relative inequalities in mortality by area deprivation consistently increased for men and women in Scotland and E&W between 1981–1983 and 2010–2012. Absolute inequalities increased for men and women in Scotland, and for women in E&W, between 1981–1983 and 2000–2002 before subsequently falling. For men in E&W, absolute inequalities were more stable until 2000–2002 before a subsequent decline. Both absolute and relative inequalities were consistently higher in men and in Scotland. The increase in absolute inequalities between 1981 and 2001 and the increase in relative inequalities between 1981 and 2011 among men was much greater in Scotland

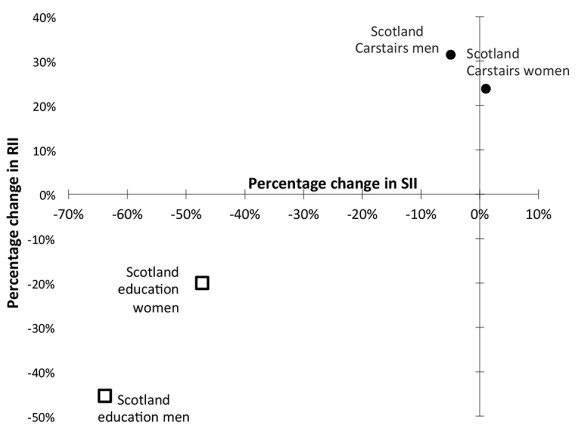

**Figure 2** Percentage change in SII and RII by Carstairs area deprivation and educational attainment (1993 to 2008, Scotland). RII, Relative Index of Inequality; SII, Slope Index of Inequality.

than in E&W. The trends in inequalities among women were more similar, although consistently higher in Scotland.

A comparison of trends for inequalities by educational attainment with those by area deprivation show highly divergent patterns. Between the early 1990s and late 2000s, Mackenbach *et al* reported large decreases in absolute and relative inequalities in Scotland, whereas the trends by area deprivation showed large increases in relative inequalities and little change in absolute inequalities for the same period. Although the data for E&W could not be compared directly, the trends in relative and absolute inequalities using deprivation do not seem to be consistent with the trends using educational attainment.

### Strengths and weaknesses
We have used data from the whole population based on area of residency to rank the population into 10 approximately equal-sized groups to calculate RIIs and SIIs for four time points. In doing so, we have been able to include the distribution of mortality across the whole population (rather than just the extremes) and have been able to avoid aggregating heterogeneous populations into large groups. Area deprivation also has the advantage of avoiding compositional change over time, in that the population can be divided and ranked easily into 10 groups at each time point, unlike educational attainment data which is subject to marked secular trends in attainment which impact differentially by age.[20] The deprivation–mortality relationship we describe in this paper is not subject to the lagged selection bias effects of some other approaches.[21]

As with all area-based measures of social ranking, the socioeconomic status applied to each individual is not a function of their personal socioeconomic status but of the mean of all those living in that area. It therefore measures a different aspect of socioeconomic status and classifies

**Table 2** RII, SII, absolute and relative differences in mortality for those aged 35–79 years by educational attainment and Carstairs deprivation

| | | Educational attainment* | | | Carstairs deprivation | | |
|---|---|---|---|---|---|---|---|
| | | 1991–1995 | 2006–2010 | % change | 1993 | 2008 | % change |
| RII | | | | | | | |
| Scotland | Men | 3.3 | 1.8 | −45 | 1.9 | 2.5 | +31 |
| Scotland | Women | 2 | 1.6 | −20 | 1.7 | 2.2 | +24 |
| SII (per 100 000 population per year) | | | | | | | |
| Scotland | Men | 1634 | 591 | −64 | 982 | 934 | −5 |
| Scotland | Women | 601 | 317 | −47 | 504 | 509 | +1 |

RII, Relative Index of Inequality; SII, Slope Index of Inequality.
*Data extracted from ref. 7.

some people who are individually socioeconomically disadvantaged as non-disadvantaged and vice versa. The measures used to create the Carstairs Deprivation Index may also be becoming a less sensitive means of characterising socioeconomic position as car ownership becomes increasingly common and overcrowding of housing less common in the UK.[22–24] It is also possible that there is some reverse causation in the relationship between area deprivation and health status (particularly in relation to younger, healthier, individuals moving to less deprived areas[25 26]). However, such population movement is unlikely to substantially undermine the results.[27–29]

### How this fits with the existing literature

The rise in health inequalities in Scotland and E&W and the causes of this are well described.[30] In particular, the rise in income inequalities and unemployment associated with the changed political context of the 1980s is likely to be a large part of any explanation.[22 28 31] It has been argued that the recent small declines in absolute inequalities for working age adults may be in part due to cohorts who were most adversely impacted by the application of neoliberalism and who were at high risk of death from alcohol-related and drug-related deaths, suicide and violence, passing through the age of highest risk.[22 32–34] In this way, there may be some lagged effects of political exposures during the 1980s and 1990s which contributed to the observed rise in health inequalities demonstrated here. Further exploration of how health inequalities have impacted on different age groups over time is merited.

The trends we describe for area deprivation here for E&W are very similar to those reported using the National Statistics Socio-Economic Classification (see online supplementary figure 1) and for Scotland using another small-area measure of multiple deprivation.[35 36] The similarity in the trends casts further doubt on the much more optimistic trends in inequalities described using educational attainment data to rank the population.[7]

Part of the reason for the markedly different trends may be the limited data available for educational attainment in Great Britain prior to 2001 which does not facilitate adequate stratification of the population to examine differences in outcomes.[14 15 37] The secular trends in educational attainment, and the differences in these trends by age group, also make it important to examine narrow age groups in any cross-sectional analysis.[10]

### Implications

Relative mortality inequalities using area deprivation in those aged 35–79 years has continuously increased since the early 1980s in Britain, and if there has been any decline in absolute inequalities since 2001, it has been small and limited to men. It is possible that there are different trends using different measures of socioeconomic status or that one or more of the measures is subject to error or bias that has given rise to the divergent trends. There are particular issues with the educational attainment data in Great Britain prior to 2001 that make these estimates less certain. Further work to find adequate means to compare the extent of inequalities internationally should be undertaken.[37]

**Contributors** GM, FP, SVK and DW came up with the idea for this work. LS and FP did the data analysis. GM drafted the manuscript. All authors provided substantial edits and redrafting and approved the final draft.

**Funding** FP and SVK are funded by the Medical Research Council and Chief Scientist's Office (MC_UU_12017/13 and SPHSU13), as part of the core funding for the MRC/CSO Social & Public Health Sciences Unit. In addition, SVK is funded by a NRS Senior Clinical Fellowship (SCAF/15/02).

**Competing interests** GM, DW and LS are employees of the National Health Service (NHS) in Scotland.

**Provenance and peer review** Not commissioned; externally peer reviewed.

**Data sharing statement** All of the available data are provided in the online supplementary appendix.

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
