## [Reviewer comments · BMJ Open]

This paper was submitted to a another journal from BMJ but declined for publication following peer review. The authors addressed the reviewers' comments and submitted the revised paper to BMJ Open. The paper was subsequently accepted for publication at BMJ Open.

ARTICLE DETAILS

TITLE (PROVISIONAL)	How do trends in mortality inequalities by deprivation and education in Scotland and England & Wales compare? A repeat cross-sectional study.
AUTHORS	McCartney, Gerry; Popham, Frank; Katikireddi, Srinivasa; Walsh, David; Schofield, Lauren

VERSION 1 - REVIEW

REVIEWER	Jon Minton University of Glasgow
REVIEW RETURNED	16-Jan-2017

GENERAL COMMENTS	This is an interesting paper with an important message about both the difference between relative and absolute measures of inequality, and also the difference between using educational outcome or carstairs as a measure of SEP. I think more could be made of the latter in light of a paper by Dowd et al which I link to below. Additional comments are below. Title: How do trends in different measures of mortality inequalities in Scotland and England & Wales compare? - this might suggest more than two measure. You've looked at a relative and an absolute measure, and I'd suggest adjusting title accordingly, to something like: How do trends in absolute and relative mortality inequalities in Scotland and England & Wales compare? Abstract: If word count available suggest putting that SII is absolute and RII is relative. Background: Two important points I think need mentioning/developing on: - An advantage of looking at RII is that it will show differences clearly in age groups where baseline mortality risk is low. If within (say) the age group 35-44 the more disadvantaged subgroup has a 20% elevated risk, a concern would be that they carry this proportionately higher risk with them as they age, which at older ages, due to higher baseline risk, translates to much higher mortality risk overall. This may be less of a concern if not interested in generational/cohort-led change, but something worth considering as a rationale for why
--

	relative measures worth keeping as well as absolute measures. - There is a strong critique about the inadequacies of educational attainment as a consistent measure of SEP over time, when the composition within groups has changed greatly, provided by Dowd et al 2014: http://ije.oxfordjournals.org/content/43/4/983.full I think the argument in this paper should be incorporated within the background and possibly the discussion section, in support of using Carstairs instead. Results: Although 5 year age bands are used within the age range 35-79 years, there still seems to be a lot of heterogeneity in terms of baseline risk and risk exposures within the age range. I would suggest including a sub-group analysis by age within supplementary material, and commented on in the main ms if they are pertinent. The age bands I would suggest for this analysis are: 35-55, 56-65, 66-79 years, though happy with alternative intervals. Something seems to be needed which looks at both working age and post-working age, and within working age at younger and older adults. I might expect RII and SII trends to be more similar at older ages, but could be wrong. Figure 1: I think these should have the confidence intervals included too. Discussion: An additional limitation appears to be not to look at the influence of cohort effects, in particular to see whether younger groups with high relative inequalities 'age into' older groups with high relative and absolute inequalities. An additional strength – going back to point in introduction – is in the use of Carstairs not education, given the methodological critique of the latter provided by Dowd. Discussion + possibly results: I think there should be more discussion of the differences in trends between Scotland and England/Wales.
--	---

REVIEWER	Paul Norman School of Geography, University of Leeds
REVIEW RETURNED	18-Jan-2017

GENERAL COMMENTS	Title: How do trends in different measures of mortality inequalities in Scotland and England & Wales compare? Manuscript Number: BMJ-2016-037169 This is a very useful paper which ably extends the literature and provides an alternative view to the education stratification used in another study. Whilst there is a 'fair' correlation between educational achievement and area deprivation, I wouldn't necessarily expect trends to act in parallel with those phenomena. Personally, I think it better to focus on area deprivation since inequalities by area are deeply entrenched (see Norman 2016). However, I note the authors (p 9) say that data are limited for education prior to 2001 but it would be feasible to have % of people in areas with a degree (albeit 10% sample but so is an input to Carstairs) and use that (though I wouldn't suggest this paper actually does this now as there is sufficient for publication in what is submitted).
---

	In terms of the methods adopted here, I agree that using SII and RII is appropriate and provides more useful outputs than simply rate ratios of least : most deprived. I would like just a little more information on the geography used at each census since it isn't clear whether the original ward / postal sector geographies have been used or whether the data have been converted to 2011 geographies and then the deprivation calculated. Having country specific deciles is interesting and the authors might like to note that a similar approach has been used before (in a study one of the co-authors was involved in) with deprivation across the whole of GB but with quintiles calculated within each GOR and Wales and Scotland (see Norman et al., 2011). A few minor edits would be useful: P.2 para 2. Measures of population health tend always to ... Which? Tend or always? Suggest just have tend. Are the deciles equal population size or equal numbers of people? Just clarify on p. 3 and p. 9. Also, as above, which time point geographies have been used (since in tandem with the cross-sectional relative calculation of deprivation and the number of geographical subdivisions and then the deciles all make a difference to the deprivation measures and then the health measures which emerge). Otherwise, well written, interesting work. Paul Norman 18th January 2017 Norman P (2016) The Changing Geography of Deprivation in Britain: 1971 to 2011 and Beyond. Chapter 11 in Champion T and Falkingham J (eds.) Population change in the United Kingdom: Rowman & Littlefield: London: 193-214 Norman P, Boyle P, Exeter D, Feng Z & Popham F (2011) Rising premature mortality in the UK s persistently deprived areas: Only a Scottish phenomenon? Social Science & Medicine 73 1575-1584 doi:10.1016/j.socscimed.2011.09.034
--	---

REVIEWER	Bjørn Heine Strand Norwegian Institute of Public Health/University of Oslo, Norway
REVIEW RETURNED	26-Jan-2017

GENERAL COMMENTS	This is an interesting paper investigating trends in mortality inequalities by area deprivation in Scotland and England & Wales (E&W). It is well written, uses sound methods and results are nicely presented and discussed. I believe the paper would interest the readers of BMJ. It is a commentary to a recent paper published in BMJ, by Mackenbach et al, which found substantial declines in absolute inequalities between high and low educational groups in Scotland and E&W between mid-1990s and mid-2000s. The submitted work questions if there indeed has been such large declines in mortality inequalities the last decades, and particularly they question the use of education as a measure of socioeconomic position (SEP) in Scotland and E&W in this period. There has been problems using education as a SEP measure in E&W because the low educated group could not be distinguished from the mid educated group in the 1991 census. Mackenbach et al acknowledge this in their paper and suggest that results for E&W should be treated with caution. However, Mackenbach noted that changes of absolute inequalities by occupational class (web table A6 in Mackenbach paper, ref 7) were also declining and thereby in support of the results regarding
--

	education. The authors of the current paper also find narrowing inequalities using the area-based SEP measure, but the declines were far more modest than those reported by Mackenbach et al using education; For E&W Mackenbach reported 36% reduction when education was used, and in the current paper the decline was 12% using the area-based SEP measure. However, using occupation there was a 20% decline for men in E&W in the Mackenbach-paper, which is more in line with the results for area-based SEP (Table A6 in the Appendix of Mackenbach-paper. Occupation based results were not reported for Scotland). SEP has several dimensions and I believe there is no surprise that education gives different results compared to aggregated area based measures. Causal pathways between education on health and mortality might differ from the pathways between aggregated area based measures and health. For example health selection effects will not be as strong regarding education (you will not lose years of education if you get sick as an adult, but you might suffer economically). This could possibly be discussed more in the paper. Table 1 and Figure 1 based on same data? Table redundant?
--	---

REVIEWER	Rampaul Chamba Deafinitions Ltd
REVIEW RETURNED	03-Feb-2017

GENERAL COMMENTS	HOW DO TRENDS IN DIFFERENT MEASURES OF MORTALITY INEQUALITIES IN SCOTLAND AND ENGLAND & WALES COMPARE? BMJ Patient Review R. Chamba 03 February 2017 FOR ALL ARTICLES (1) Is the article important? Yes because it contributes to an examination of how different measures may yield different trends or error and bias; it highlights the strengths and limitations of different measures; and it contributes to the role of socio-economic and educational variables in explaining inequalities. (2) Will it help our readers to make better decisions and, if so, how? I'm not sure which readers are being referred to. The paper may help readers interested in methodology and the strengths and limitations of different measures to be more discerning about how they pursue similar future lines of research. Given the different trends observed between educational attainment and area deprivation it is unclear how these findings could begin to inform policy or practice. (3) Will the article add enough to existing knowledge? It's an interesting contribution. (4) Does the article read well and make sense? Does it have a clear message? The article is well written but there's also a lot of information packed
---

	into some sentences. For e.g., on page 9/line 30-34 starting with "It has been argued..." could do with expanding and elaboration. FOR PATIENT REVIEWERS (1) Are the study's aims and the issue and questions that the paper addresses relevant and important to you as a patient? Do you think it would be relevant to other patients like you? What about carers? Not particularly. As a patient and in terms of the issues that impinge on my experience of health services and health inequalities, my concern is probably more with how services on the ground could be made better to improve access rather than the nuances of which measure provides a better explanation of apparent trends in health inequalities. (2) Are there any areas that you find relevant as a patient or carer that are missing or should be highlighted? Given the intended remit of the paper, I'd be interested to know how the conclusions of the paper align with any qualitative studies about patients'/carers' experiences of services in Scotland and E&W and how they perceive the importance of educational attainment and area deprivation on health inequalities. A brief consideration of this from a qualitative perspective would have been helpful and some references. (3) From your perspective as a patient, would the treatment, intervention studied, or guidance given actually work in practice? Is it feasible? What challenges might patients face that should be considered? Pass. (4) Are the outcomes that are being measured in the study or described in the paper the same as the outcomes that are important to you as a patient? Are there others that should have been considered? See response to (2) above. (5) Do you have any suggestions that might help the author(s) strengthen their paper to make it more useful for doctors to share and discuss with patients? See response to (2) above. (6) The level of patient involvement in the research described, and if and how it could have been improved. Authors are now required to state if and how they involved patients in setting the research agenda and the design and implementation of the study and include this information in a box within the manuscript. If there was no patient involvement we would welcome your ideas on how this could have been done. We hope this will help authors think of the best ways to include patients in their future research and further progressive patient involvement in the research enterprise. See response to (2) above.
--	--

VERSION 1 – AUTHOR RESPONSE

Response to reviewers

Reviewer: 1 (Jon Minton)

This is an interesting paper with an important message about both the difference between relative and absolute measures of inequality, and also the difference between using educational outcome or carstairs as a measure of SEP. I think more could be made of the latter in light of a paper by Dowd et al which I link to below.

Title:

How do trends in different measures of mortality inequalities in Scotland and England & Wales compare? - this might suggest more than two measure. You've looked at a relative and an absolute measure, and I'd suggest adjusting title accordingly, to something like:

How do trends in absolute and relative mortality inequalities in Scotland and England & Wales compare?

Our response > We have amended the title to make it clear that the principle comparison being made here is the use of area deprivation to rank the population with the use of educational attainment to rank the population.

Abstract:

If word count available suggest putting that SII is absolute and RII is relative.

Our response > We have added this into the abstract.

Background:

Two important points I think need mentioning/developing on:

- An advantage of looking at RII is that it will show differences clearly in age groups where baseline mortality risk is low. If within (say) the age group 35-44 the more disadvantaged subgroup has a 20% elevated risk, a concern would be that they carry this proportionately higher risk with them as they age, which at older ages, due to higher baseline risk, translates to much higher mortality risk overall. This may be less of a concern if not interested in generational/cohort-led change, but something worth considering as a rationale for why relative measures worth keeping as well as absolute measures.

Our response > The patterning of inequalities in mortality at different ages is an important and interesting question, but it would represent a substantial extension of the analysis and would move the focus of the paper away from a comparison with the method used by Mackenbach et al. We do not therefore propose adding age-stratified analysis to this paper.

- There is a strong critique about the inadequacies of educational attainment as a consistent measure of SEP over time, when the composition within groups has changed greatly, provided by Dowd et al 2014:

<http://ije.oxfordjournals.org/content/43/4/983.full>

I think the argument in this paper should be incorporated within the background and possibly the discussion section, in support of using Carstairs instead.

Our response > This is an important issue and a limitation of methods that use a measure of socioeconomic position that is subject to a secular trend and where lagged selection biases are likely. The relative merits of an approach based on the area of residence compared to one which uses educational attainment, and the implications in terms of secular trends is already discussed. For both the deprivation analysis we present, there is little or no time lag in the measurement of exposure and outcome. We have added the reference to Dowd and made this point clearer in the discussion.

Results:

Although 5 year age bands are used within the age range 35-79 years, there still seems to be a lot of heterogeneity in terms of baseline risk and risk exposures within the age range. I would suggest including a sub-group analysis by age within supplementary material, and commented on in the main ms if they are pertinent. The age bands I would suggest for this analysis are: 35-55, 56-65, 66-79 years, though happy with alternative

intervals. Something seems to be needed which looks at both working age and post-working age, and within working age at younger and older adults. I might expect RII and SII trends to be more similar at older ages, but could be wrong.

Our response > This is a useful suggestion but would require substantial additional analysis which is outwith the scope of this paper, the prime aim of which is to directly compare trends in mortality inequality using deprivation with those using educational attainment.

Figure 1: I think these should have the confidence intervals included too.

Our response > We have now added these into the chart and the supplement.

Discussion:

An additional limitation appears to be not to look at the influence of cohort effects, in particular to see whether younger groups with high relative inequalities 'age into' older groups with high relative and absolute inequalities.

Our response > As noted above, the purpose of this paper was to compare mortality inequalities by area deprivation directly with the published mortality inequalities by educational attainment. As such, the we age standardised rather than age stratified to ensure comparability. The suggestion to look at age and cohort effects is a good one, but beyond the scope of this paper.

An additional strength – going back to point in introduction – is in the use of Carstairs not education, given the methodological critique of the latter provided by Dowd.

Our response > This strength has now been clarified in the discussion.

Discussion + possibly results:

I think there should be more discussion of the differences in trends between Scotland and England/Wales.

Our response > We have added a little more discussion of this in first part of the discussion.

Reviewer: 2 – Paul Norman

This is a very useful paper which ably extends the literature and provides an alternative view to the education stratification used in another study. Whilst there is a 'fair' correlation between educational achievement and area deprivation, I wouldn't necessarily expect trends to act in parallel with those phenomena. Personally, I think it better to focus on area deprivation since inequalities by area are deeply entrenched (see Norman 2016). However, I note the authors (p 9) say that data are limited for education prior to 2001 but it would be feasible to have % of people in areas with a degree (albeit 10% sample but so is an input to Carstairs) and use that (though I wouldn't suggest this paper actually does this now as there is sufficient for publication in what is submitted). In terms of the methods adopted here, I agree that using SII and RII is appropriate and provides more useful outputs than simply rate ratios of least : most deprived. I would like just a little more information on the geography used at each census since it isn't clear whether the original ward / postal sector geographies have been used or whether the data have been converted to 2011 geographies and then the deprivation calculated. Having country specific deciles is interesting and the authors might like to note that a similar approach has been used before (in a study one of the co-authors was involved in) with deprivation across the whole of GB but with quintiles calculated within each GOR and Wales and Scotland (see Norman et al., 2011).

Our response > We thank the reviewer for his comments. We have added a sentence in the methods to clarify our use of geographies as requested.

A few minor edits would be useful:

P.2 para 2. Measures of population health tend always to ... Which? Tend or always? Suggest just have tend.

Are the deciles equal population size or equal numbers of people? Just clarify on p. 3 and p. 9.

Also, as above, which time point geographies have been used (since in tandem with the cross-sectional relative calculation of deprivation and the number of geographical subdivisions and then the deciles all make a difference to the deprivation measures and then the health measures which emerge).

Our response > We have added clarification on each of these points to the manuscript.

Otherwise, well written, interesting work.

Reviewer: 3 - Bjørn Heine Strand

This is an interesting paper investigating trends in mortality inequalities by area deprivation in Scotland and England & Wales (E&W). It is well written, uses sound methods and results are nicely presented and discussed. I believe the paper would interest the readers of BMJ.

It is a commentary to a recent paper published in BMJ, by Mackenbach et al, which found substantial declines in absolute inequalities between high and low educational groups in Scotland and E&W between mid-1990s and mid-2000s. The submitted work questions if there indeed has been such large declines in mortality inequalities the last decades, and particularly they question the use of education as a measure of socioeconomic position (SEP) in Scotland and E&W in this period. There has been problems using education as a SEP measure in E&W because the low educated group could not be distinguished from the mid educated group in the 1991 census. Mackenbach et al acknowledge this in their paper and suggest that results for E&W should be treated with caution. However, Mackenbach noted that changes of absolute inequalities by occupational class (web table A6 in Mackenbach paper, ref 7) were also declining and thereby in support of the results regarding education.

The authors of the current paper also find narrowing inequalities using the area-based SEP measure, but the declines were far more modest than those reported by Mackenbach et al using education; For E&W Mackenbach reported 36% reduction when education was used, and in the current paper the decline was 12% using the area-based SEP measure. However, using occupation there was a 20% decline for men in E&W in the Mackenbach-paper, which is more in line with the results for area-based SEP (Table A6 in the Appendix of Mackenbach-paper. Occupation based results were not reported for Scotland).

SEP has several dimensions and I believe there is no surprise that education gives different results compared to aggregated area based measures. Causal pathways between education on health and mortality might differ from the pathways between aggregated area based measures and health. For example health selection effects will not be as strong regarding education (you will not lose years of education if you get sick as an adult, but you might suffer economically). This could possibly be discussed more in the paper.

Our response > We agree with most of the reviewer's comments here, although the lack of data presented for Scotland on occupational group by Mackenbach et al limited our ability to compare on this measure. We have added a sentence on the possibility of selection effects in the discussion.

Table 1 and Figure 1 based on same data? Table redundant?

Our response > This is the same data presented in the table and chart. We would be happy to take editorial advice on whether to include both or not.

Reviewer: 4 – (BMJ Patient Review) R. Chamba

(1) Is the article important?

Yes because it contributes to an examination of how different measures may yield different trends or error and bias; it highlights the strengths and limitations of different measures; and it contributes to the role of socio-economic and educational variables in explaining inequalities.

(2) Will it help our readers to make better decisions and, if so, how?

I'm not sure which readers are being referred to. The paper may help readers interested in methodology and the strengths and limitations of different measures to be more discerning about how they pursue similar future lines of research. Given the different trends observed between educational attainment and area deprivation it is unclear how these findings could begin to inform policy or practice.

Our response > The policy implications of the paper concern whether or not (and to what degree) health inequalities are improving. This will influence the priority given to policies which are designed to address health inequalities and the understanding of the underlying mechanisms.

(3) Will the article add enough to existing knowledge?

It's an interesting contribution.

(4) Does the article read well and make sense? Does it have a clear message?

The article is well written but there's also a lot of information packed into some sentences. For e.g., on page 9/line 30-34 starting with "It has been argued..." could do with expanding and elaboration.

Our response > We have added further elaboration of this point as requested.

FOR PATIENT REVIEWERS

(1) Are the study's aims and the issue and questions that the paper addresses relevant and important to you as a patient? Do you think it would be relevant to other patients like you? What about carers?

Not particularly. As a patient and in terms of the issues that impinge on my experience of health services and health inequalities, my concern is probably more with how services on the ground could be made better to improve access rather than the nuances of which measure provides a better explanation of apparent trends in health inequalities.

(2) Are there any areas that you find relevant as a patient or carer that are missing or should be highlighted?

Given the intended remit of the paper, I'd be interested to know how the conclusions of the paper align with any qualitative studies about patients'/carers' experiences of services in Scotland and E&W and how they perceive the importance of educational attainment and area deprivation on health inequalities. A brief consideration of this from a qualitative perspective would have been helpful and some references.

(3) From your perspective as a patient, would the treatment, intervention studied, or guidance given actually work in practice? Is it feasible? What challenges might patients face that should be considered?

Pass.

(4) Are the outcomes that are being measured in the study or described in the paper the same as the outcomes that are important to you as a patient? Are there others that should have been considered?

See response to (2) above.

(5) Do you have any suggestions that might help the author(s) strengthen their paper to make it more useful for doctors to share and discuss with patients?
See response to (2) above.

(6) The level of patient involvement in the research described, and if and how it could have been improved. Authors are now required to state if and how they involved patients in setting the research agenda and the design and implementation of the study and include this information in a box within the manuscript. If there was no patient involvement we would welcome your ideas on how this could have been done. We hope this will help authors think of the best ways to include patients in their future research and further progressive patient involvement in the research enterprise.
See response to (2) above.

VERSION 2 – REVIEW

REVIEWER	Jon Minton University of Glasgow, Scotland, UK
REVIEW RETURNED	22-May-2017

GENERAL COMMENTS	I think the authors for making the changes suggested. I am happy with this revision. (Note a right parenthesis is missing at the point marked with a * in the new abstract sentence: "Absolute inequalities (measured using the Slope Index of Inequality (SII)) and relative inequalities (measured using the Relative Index of Inequality (RII)* in all-cause mortality.")
--

REVIEWER	Paul Norman School of Geography University of Leeds
REVIEW RETURNED	23-May-2017

GENERAL COMMENTS	Thanks for submitting this paper and for using the reviews for the original BMJ submission to improve this already good paper. Seeing the other reviews and responses has been very useful. A few minor elements: p. 5 first paragraph of statistical analysis. It would be better to say "... were constrained to the revised mid-year estimates ..." (because nothing is uniform and adjustments for differential enumeration (rather than simply undercount) is not always up (by age and sex). Sorry I didn't spot this before. p. 8 second paragraph addition of last sentence from the first submission re reverse causation and movements. Unfortunately, this is lacking in enough coverage of the health-deprivation-migration inter-relationships. Mostly, it is found that health-selective migration at least maintains if not exaggerates the health-deprivation relationship though there is some amelioration due to movements by the elderly (Norman et al., 2005). Relevant to the ages in this paper, especially in mid-life, migration between differently deprived areas leads to inequalities (Norman & Boyle, 2014). Poor health leading to movement is generally about the elderly (so yes, unlikely to be substantial) but good health enabling movement is relevant and most likely to maintain / exaggerate the inequalities. If the authors
--

	want to retain the sentence they have then this needs to be preceded by something which notes that health selective migration may be important. Norman P, Boyle P & Rees P (2005) Selective migration, health and deprivation: a longitudinal analysis. Social Science & Medicine 60(12): 2755-2771 Norman P & Boyle P (2014) Are health inequalities between differently deprived areas evident at different ages? A longitudinal study of census records in England & Wales, 1991-2001. Health & Place 26:88-93 http://dx.doi.org/10.1016/j.healthplace.2013.12.010
--	--

VERSION 2 – AUTHOR RESPONSE

Reviewer: 1 - Jon Minton

I think the authors for making the changes suggested. I am happy with this revision. (Note a right parenthesis is missing at the point marked with a * in the new abstract sentence: "Absolute inequalities (measured using the Slope Index of Inequality (SII)) and relative inequalities (measured using the Relative Index of Inequality (RII)* in all-cause mortality.")

– Our response: Thank you. The missing parenthesis has been corrected.

Reviewer: 2 - Paul Norman

Thanks for submitting this paper and for using the reviews for the original BMJ submission to improve this already good paper. Seeing the other reviews and responses has been very useful.

A few minor elements:

p. 5 first paragraph of statistical analysis. It would be better to say "... were constrained to the revised mid-year estimates ..." (because nothing is uniform and adjustments for differential enumeration (rather than simply undercount) is not always up (by age and sex). Sorry I didn't spot this before.

– Our response: we have clarified this sentence as suggested

p. 8 second paragraph addition of last sentence from the first submission re reverse causation and movements. Unfortunately, this is lacking in enough coverage of the health-deprivation-migration inter-relationships. Mostly, it is found that health-selective migration at least maintains if not exaggerates the health-deprivation relationship though there is some amelioration due to movements by the elderly (Norman et al., 2005). Relevant to the ages in this paper, especially in mid-life, migration between differently deprived areas leads to inequalities (Norman & Boyle, 2014). Poor health leading to movement is generally about the elderly (so yes, unlikely to be substantial) but good health enabling movement is relevant and most likely to maintain / exaggerate the inequalities. If the authors want to retain the sentence they have then this needs to be preceded by something which notes that health selective migration may be important.

– Our response: this is a helpful comment. We have redrafted this section and referenced the suggested papers to reflect this more clearly, as indicated below:

"It is also possible that there is some reverse causation in the relationship between area deprivation and health status as a consequence of migration [25, 26]). For example, healthier working age individuals may move to less deprived areas which may in turn lead to increasing socioeconomic inequalities in health, as assessed by area deprivation. However, previous work suggests such population movement is unlikely to substantially undermine the results.[27, 28][29]"

We hope this addresses the comments from the reviewers.

VERSION 3 - REVIEW

REVIEWER	Paul Norman School of Geography, University of Leeds
REVIEW RETURNED	07-Jun-2017

GENERAL COMMENTS	Thanks for making these minor revisions. The paper reads very well and I look forward to seeing it in 'print'.
--